# Hidden Costs Associated with Smallholder Family-Based Broiler Production: Accounting for the Intangibles

**Rafael Araujo Nacimento** [1,*], **Mario Duarte Canever** [2], **Cecilia Almeida** [3], **Feni Agostinho** [3], **Augusto Hauber Gameiro** [1] and **Biagio Fernando Giannetti** [3]

[1] Department of Animal Science, University of São Paulo, São Paulo 05508-220, Brazil; gameiro@usp.br
[2] Agrarian Social Science Department, Federal University of Pelotas, Pelotas 96015-700, Brazil; canever@ufpel.edu.br
[3] Graduation Program on Production Engineering, Paulista University, São Paulo 04026-002, Brazil; feni@unip.br (F.A.); biafgian@unip.br (B.F.G.)
[*] Correspondence: rafael.nacimento@usp.br

**Abstract:** The contractual relationship between the processing firm and the broiler smallholder presents incessant conflicts of interest and inequality due to technical and economic discrepancies, leading to an undervaluation of the producers' remuneration. This study aims to deepen the discussion on searching for a more balanced monetary exchange between processing firms and broiler smallholders based on scientific aspects. For this, the emergy theory and its concepts are used while considering a representative broiler production system at Concórdia, Brazil. The results indicate the importance of including cultural information in the emergy-based model calculation, which achieved the highest emergy contribution (~63%; transformity = $1.73 \times 10^8$ sej/J) for the broiler smallholder. On the other hand, the cultural information was not sufficient to increase the sustainability of the broiler production system. The results show an imbalance in the monetary exchange between the processing firm and broiler smallholder from both perspectives (the economic and emergy-based ones), which indicates higher values (USD 0.32/broiler and EmUSD 1.62/broiler) than the practiced payment value of USD 0.24/broiler. Evaluating the "(eco)cost" from an emergy-based accounting perspective recognizes that production depends not only on tangible physical resources but also on knowledge, skills and information ("iceberg of value" thinking). Policy and decision makers must therefore consider the promotion of public policies that subside initiatives, including social and environmental welfare programs.

**Keywords:** agricultural policy; emergy; environmental assessment

## 1. Introduction

Poultry farming is an activity that generates important economic dividends all over the world. It contributes to food security in several developing countries, in addition to generating jobs, income and foreign exchange [1]. Brazil is one of the countries with the lowest production costs for poultry production [2], which provides competitive advantages and places the country among the world's largest producers. National production jumped from 375,000 tons in 1975 to 14.25 million tons in 2022, growth of approximately 38 times [3], which is unusual among the main agribusiness activities in the country. In 2022, the Brazilian poultry chain made available 4.1 million direct and indirect jobs, aside from employing ~50,000 broiler smallholder families. In addition, 13.5% of the GDP of Brazilian agribusiness in 2022 was a result of the poultry production chain, earning USD 21.7 billion [4].

The origin of commercial poultry farming in Brazil dates back to the first half of the last century. During that time, the production was located mainly in the southeastern region of the country. However, from the 1970s onward, poultry farming became more technical and advanced toward the south of Brazil, with the incorporation of new productive and

organizational structures [5]. The supply of grains (corn and soybeans), subsidized credit and the imported genetic and health technologies stood out among the driving factors. Another additional factor for the development of poultry farming in the southern portion of the country was the adoption of a production model based on the integration between processing firms and smallholder families [6,7]. This strategy was inspired by the integrated model of chicken production already implemented in the United States, known as the Southern Model [8]. In this model, the processing firm (slaughterhouse) does not need to produce everything internally or, on the contrary, buy everything from independent external suppliers. Instead, it organizes the creation of productive integration among a set of partners (rural producers).

The smallholder families (called "integrated" families) became responsible for raising the chickens and receiving the main inputs from the industry, such as feed, medicines and chicks, and technical assistance. In this model, the chickens are delivered to the processing firm in exchange for remuneration for the work and goods demanded during poultry growth. This integrated production model is currently the basis for Brazilian poultry's competitiveness as it provides transactional efficiency (low production costs) to the system [5].

The relationship between the processing firm and the smallholder family faces challenges. The contractual relationship is notably unequal due to the technical and economic discrepancies between industrial conglomerates (oligopsony) and small farmers [9,10]. Thus, the structure of this market tends to undervalue the remuneration of producers [11,12]. In general, contracts place the broiler smallholders in direct dependence on the economic and political power of processing firms, removing them from the free competitive market and making them co-adjuvants in the establishment of product prices offered to the processing firms [5,13]. It was in this context and after many demands from the integrated producers that integration law was created in Brazil in 2016 (Federal Law n.° 13,288), which came to standardize the relationship between the processing firm and broiler smallholder. Although important, the law does not state how the remuneration of the integrated member should be handled. It only imposes new mechanisms to alleviate the asymmetries in the relationship through the creation of a bipartite commission of representatives of the integrated members and integrating companies [10].

Broiler smallholder remuneration methods are generally based on productive efficiency indices [14] and economic cost indicators [15]. As processing firms have more power than integrated farmers, it is highly likely that the prices received by the farmer (weaker party) are depressed. Economic theory [16] shows that an agent should only remain in a given activity when what he or she receives covers all expenses with that activity. Therefore, for an integrated poultry farm, its remuneration should cover all costs associated with the operation (such as labor, energy and materials), known as explicit costs. However, the mainstream economic costs consider neither the expenses with the so-called free factors (such as the energy of sunlight and water) nor the contribution of the cultural information brought by an indigenous population to an agricultural production system. As shown by Abel [17,18], cultural knowledge plays an important role in sustaining the progress of society and agricultural systems.

In the case of the integrated poultry production system in southern Brazil, the cultural contribution of family farmers to the success of the business model is notorious. Family farmers are descendants of European settlers who exalted the value of work, the cult of discipline, the importance of community and family life [19]. These values shaped family farming and were fundamental to the successful integration of family smallholders with processing firms over time [5,19]. Therefore, cultural information provides support to the agricultural production systems [20–22] and is part of the total contribution of the poultry farmers to the integrated production system.

For measuring the total contribution of farmers to the poultry system, it is inadequate to use the classical economic approach. This discussion needs to be supported by a holistic point of view based on the total effort of smallholders and the nature of the production

system. For this purpose, "iceberg (or pyramid) of value" thinking could help to deepen the discussion regarding the smallholder payment and consider a holistic point of view and system thinking. In the "iceberg (or pyramid) of value" perspective, the market and cost values are observed from the mainstream economic system, whereas there are hidden resources from the "visible peak" supporting the production system, but they are not measured from the mainstream economic metrics. To measure the hidden costs from a holistic point of view, it is appropriate to use a holistic tool such as emergy [23].

Emergy is a theory and method that quantifies all energy inputs (embodied energy) needed to generate some product or service [24–26]. Emergy synthesis can put together all environmental, economic, and sociocultural efforts provided by the smallholder families to the broiler production chain in a unique common base (solar-equivalent energy). Converting this total flow into a monetary-equivalent payment is a way to quantitatively bring the intangibles (cultural information and other hidden costs) into economic indicators. According to Abel [18], emergy synthesis is a tool for characterizing the system structure, namely whether or not it is self-organized with information control. Where information does exist, emergy is a method for judging the relative impact of its control. It is a means to "locate" its production within the hierarchy of energy transformation processes [18]. Thus, measuring all energy-matter inputs used in broiler production in a common base (solar-equivalent energy) and then converting the flow to a monetary-equivalent payment could be a way to quantitatively bring the intangibles into economic indicators and search for a holistic view-based payment for the smallholder broiler family through their services.

This article aims to deepen the discussion on searching for a more balanced monetary exchange between processing firms and broiler smallholders based on scientific aspects from a holistic perspective using emergy theory. Aside from providing discussions based on quantitative data regarding the broiler production case in Brazil, this work contributes to discussions about the usage of emergy synthesis as an alternative in measuring the value of intangible resources, an important and current debate toward the Agenda 2030 goals.

## 2. Contextualizing the Importance of Intangibles in Sustaining Smallholder Family-Based Broiler Production

Most of the south Brazilian land occupation took place from the second decade of the 19th century onward and was carried out by descendants of European immigrants [27]. From the beginning, raising animals (mainly pigs) was one of the ways to pay for land and was of paramount importance for the development of the region. Therefore, the knowledge and ability to deal with small animals has been one of the characteristics of these farmers since the beginning of territorial occupation [5,19]. From small factories (mostly salami and lard), the enterprises diversified into poultry production from the 1970s onward. Since its origin, poultry farming in the region has been developing from the smallholder family and processing firm relationship via integration contracts. Integration provided control mechanisms over the production process, with family farming as the social basis for its support.

As the settler strongly envisaged social ascension, the arrival of poultry farming was seen as an opportunity, including the advantages of using the workforce of women and children in daily operations. Farmers saw this as another opportunity to increase their family income. Poultry farming is a confined production which requires specialized labor and exclusive dedication to the care of the animals. It is necessary to be permanently present in poultry facilities even at night or on weekends, which requires obedience to the rules established in the contract [10].

The integrating company guarantees the availability of raw material in a sufficient quantity and quality for production without the need to incur huge production costs if it adopts the verticalization strategy for production. On the other hand, the integration contract reduces transaction costs both ex ante (to find farmers, to negotiate prices and to draft contractual instruments) and ex post (to enforce the obligations assumed). Therefore, the role played by the farmer through contracting with the processing firm has been fundamental to the performance of the Brazilian poultry industry, which has grown by

more than 38 times in terms of volume produced from the mid-1970s until today. However, despite its unquestionable contributions, the knowledge and cultural information of agricultural smallholders embodied within poultry production are not fully evaluated in this market [18].

Cultural information is turned into techniques that support agricultural production [20–22], improving production and food security (Figure 1). However, since the information is always "carried" on a material or energetic carrier of some kind, to maintain the "information carrier", the human controller is primordial [18]. When it comes to humans, this maintenance requires satisfactory levels of social welfare and good quality of life to support it. According to Odum [25], higher welfare levels for humans are obtained when there is a good match between the small-, medium- and large-scale inputs. In this sense, environmental influence, exergy-level support (i.e., fuel, electricity and transport) and information exchange are inputs that may be required for human individual benefits. According to Brandt-Williams [28], wage and food input flows are needed to produce work. Additionally, the information generated for this human feeds back into the knowledge network as an autocatalytic process.

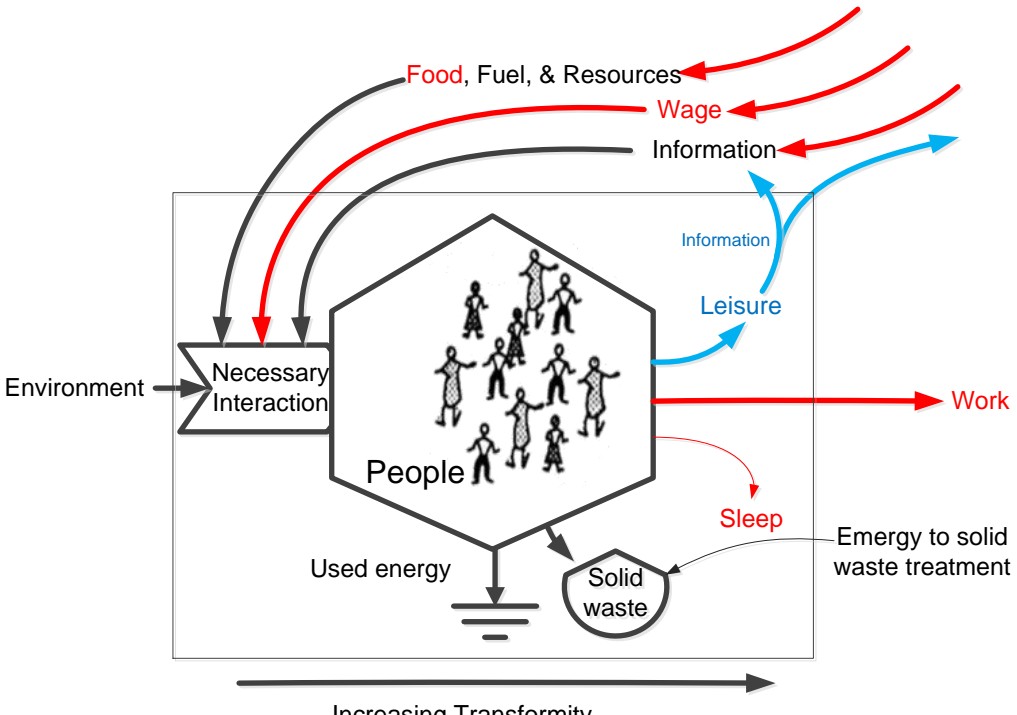

**Figure 1.** Aggregated emergy diagram of a citizen with emergy requirements for the welfare of human individuals, including fuels, resources (e.g., minerals and services), environment and information [25] (in black) as well as emergy flows of worker, including food and wage [28] (in red), and work, with sleep as output. Information feedback can be generated in leisure time (in blue).

## 3. Methods

### 3.1. Mainstream Economic Cost Model

The economic cost model was developed according to economic theory, considering the total cost as the result of the sum of the variable cost and the fixed cost (total cost = variable cost + fixed cost) and following the scheme proposed by Nacimento et al. [29] (Figure 2). The calculation procedure is provided in Table S1 of Supplementary Material A.

| Variable costs | | Fixed costs |
|---|---|---|
| Electric power | | Production cost factor* |
| Heating | | Litter |
| Fuel | Smallholder family | Tax |
| Insurance | | Depreciation |
| Miscellaneous | | Maintenance |
| Registered mapower | | Cleaning & Sanitization |
| Transport | | |
| One-day-chicks | | |
| Catching services | Processing firm | |
| Technical assistance | | |
| Nutrition | | |
| Health care | | |

**Figure 2.** Scheme representing the division of inputs for cost assessment in a conventional broiler production system. The inputs are divided according to their respective partners in variable and fixed costs. Note: the dashed line square shows the inputs under smallholder responsibility, and the continuous line square shows the inputs under processing firm responsibility. Despite some particularities among broiler processing firms, this is the most common division of responsibilities. * Production cost factor = opportunity cost of working capital, land, facilities and equipment use.

### 3.2. Alternative Economic Cost Model: Emergy Synthesis as a Way to Include the Intangibles

Emergy theory was proposed by Odum [25,26] as being the whole energy previously needed to produce goods and services. The unit of emergy is the solar emjoule (sej), and emergy synthesis considers the effort of nature in human production processes. The methodology depicts the systems' features, including their driving forces and interactions [30]. The emergy synthesis applied in this study consists of three steps: (1) elaborating a diagram of the energy flows of the system, defining the energy sources, the system boundaries and the internal components (producers, consumers, stocks, interactions, etc.); (2) organization and elaboration of tables for calculating the emergy flows, multiplying the transformities by the energy or material flow of each input, and (3) calculating the emergy indices that support discussion of the system's emergy performance. The demands for local renewable (R) and non-renewable (N) resources are included (I = R + N), as well as inputs from the economy (F), which are divided into renewable ($F_R$) and non-renewable ($F_N$) fractions of each source. The sum of these inputs (Y = I + F) indicates the total emergy demanded by the systems.

According to Odum [25,26], transformity (Tr) is a measure of the hierarchy of energy, be it matter or information. As a concept, the Tr refers to the energy quality of a given product or service, whilst as an indicator, the Tr refers to the energy conversion efficiency of the production system. For the purpose of this study, the Tr values of the items listed in the calculation table were obtained from the scientific literature and, when necessary, corrected to the emergy baseline proposed by Brown et al. [31] of $12.0 \times 10^{24}$ seJ/J.

As a way of incorporating the important variables (intangibles) of smallholder family-based broiler production into economic data, the following three approaches were considered: (1) estimation of cultural information transformity based on emergy theory [25]; (2) transformity assessment for the smallholder broiler family labor force with and without cultural information and (3) estimation of a more balanced payment for the smallholder broiler family. All of these approaches are detailed and separately presented in the following sections.

### 3.2.1. Estimating the Transformity for Cultural Information

According to Odum, information consists of the unit, connections and configurations of systems. Also, information is responsible for carrying material energy of some kind [18]. Information carriers are subject to the second law of thermodynamics (entropy) and must be maintained with "information cycles" that comprehend selection, extraction, copying and dispersal [18]. The information has the functional activities (1) to maintain time-tested energy pathways and processes and (2) to provide control in system designs that (self-)organize for maximum empower [18,26].

Previous studies used emergy theory to evaluate the ways of sharing information as well as its influence on education and culture [18]. However, although such information was a frequent component of computer simulations in emergy synthesis, it was often treated as a peripheral element in larger studies [18].

According to Abel [18], any information transformities could only be calculated under area or populational analyses. Thus, the method could provide an "areal" and "populational" calculation of information transformities, specifically considering the human body, information and DNA creation and maintenance concerning energy storage and flow. In this sense, the evaluation of storage is intended to represent the original production of information, whereas the evaluation of flow is intended to represent the maintenance of information.

In this study, the method used to estimate the Tr for the cultural information of Catarina State citizens followed an emergy-based model proposed by Odum and Doherty [32] and Odum [25]. The Tr was obtained by dividing the emergy flow of cultural information by the energy storage in Santa Catarina citizens. The emergy flow of cultural information was obtained multiplying the emergy flow of the renewable resources of Santa Catarina, Brazil (in solar emjoules (sej) [33]) by 100 years. According to the scientific literature, there is the understanding that the social interactions among indigenous people suffered over the past ~100 years. The earliest and newest European immigrant people were responsible for promoting the needed knowledge exchange to push and strengthen the broiler industry in the western portion of Santa Catarina, making cultural information an important control flow [34,35]. The energy storage in Santa Catarina citizens was obtained by considering 10% of human metabolism for social interaction and learned information (in joules (J)) [25]. The populational information of Santa Catarina, Brazil, for 2018, was found according to the Brazilian Institute of Geography and Statistics [36] (Table S2; Supplementary Material B, the "Cultural information" sheet).

### 3.2.2. Calculating the Transformity of the Broiler Smallholder Service with and without Cultural Information

To determine the Tr of the work from the broiler smallholder, "areal" and "populational" data were recorded from governmental agency reports for the city of Concórdia, Brazil, following the steps suggested by Su et al. [37]. The city of Concórdia was selected due to its current and historical importance in Brazilian broiler production, being the cradle of modern Brazilian broiler production. The reports compiled the data of forests and water bodies, agricultural lands and uncovered areas, and import and export trade data were considered to analyze the emergy flows of urban ecosystems (Figure 3).

The renewable resources flowing into the urban ecosystem included sunlight, rain (geopotential energy and chemical potential energy), wind and Earth cycles. The non-renewable resources were associated with the support of citizens to assure social welfare. According to Odum [25], the three kinds of inputs of different transformities that may be required for human individual benefit (humane welfare) are as follows: (1) environmental influence, (2) exergy-level support (available energy of medium transformity sources such as fuels and electricity) and (3) informational exchange. Waste may act as a negative influence. On the other hand, in modern society, humans provide their workforce as the main output and receive monetary payment [28]. Thus, the non-renewable resources included food, hydroelectric power, fuel (ethanol, gasoline and lubricants), natural gas, information,

wages and solid waste treatment (Figure 4; for details, see Supplementary Material B, the "Manpower" sheet) (Table S3).

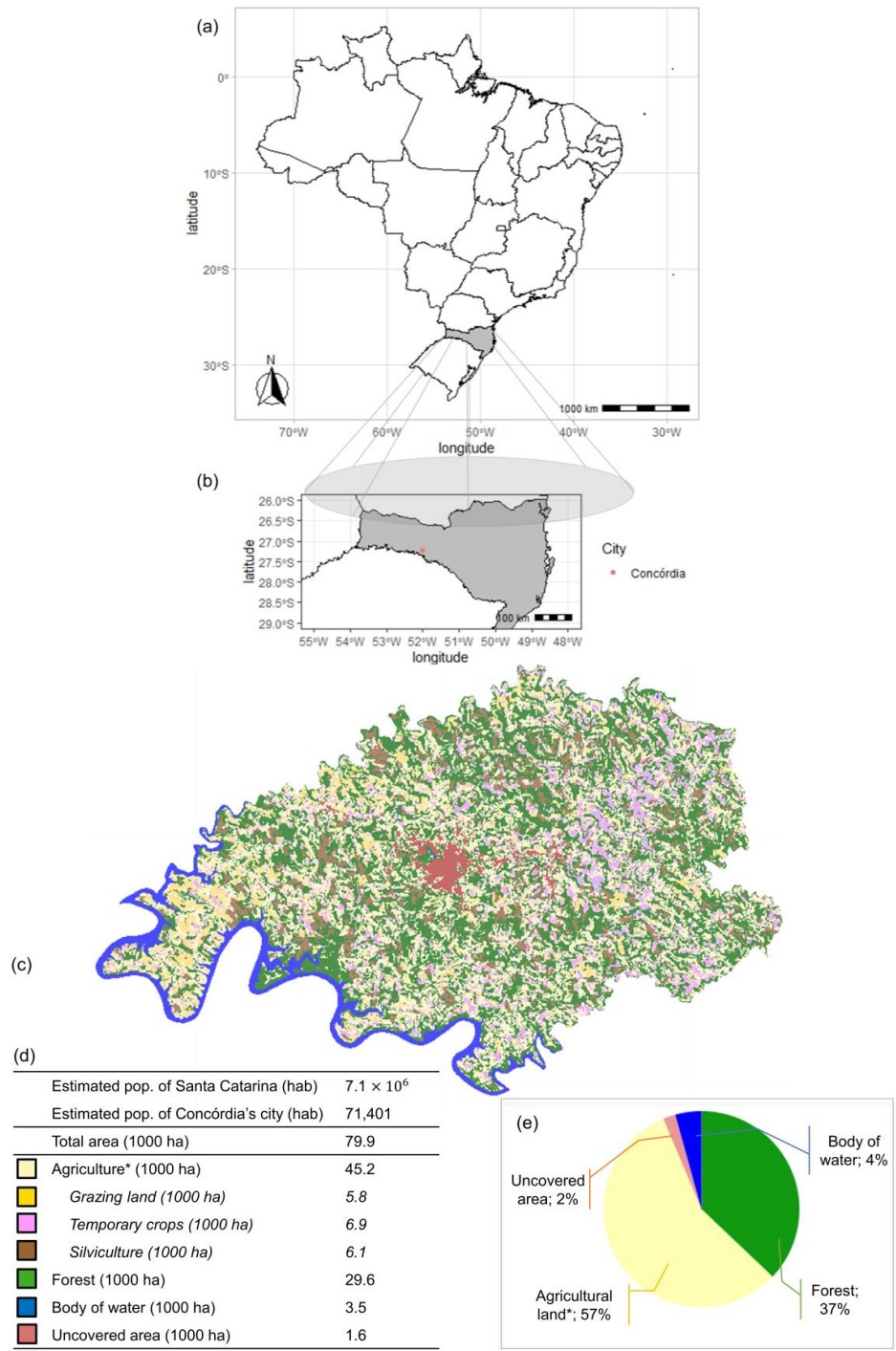

**Figure 3.** Location of Concórdia (Santa Catarina, Brazil), with area and population information for 2018. (**a**) Brazilian map and location of Santa Catarina. (**b**) Location of Concórdia (pink point). (**c**) Land cover and landscape distribution of Concórdia. (**d**) Numerical information regarding land cover and landscape distribution of Concórdia; (**e**) Percentage of the Concórdia landscape distribution. Source: (**a**,**b**) available at https://www.gps-coordinates.net/, accessed on 18 July 2023 (for latitude and longitude data); (**c**–**e**) available at https://mapbiomas.org/, accessed on 18 July 2023. * Considering agricultural land as the sum of grazing land, temporary crops and silviculture.

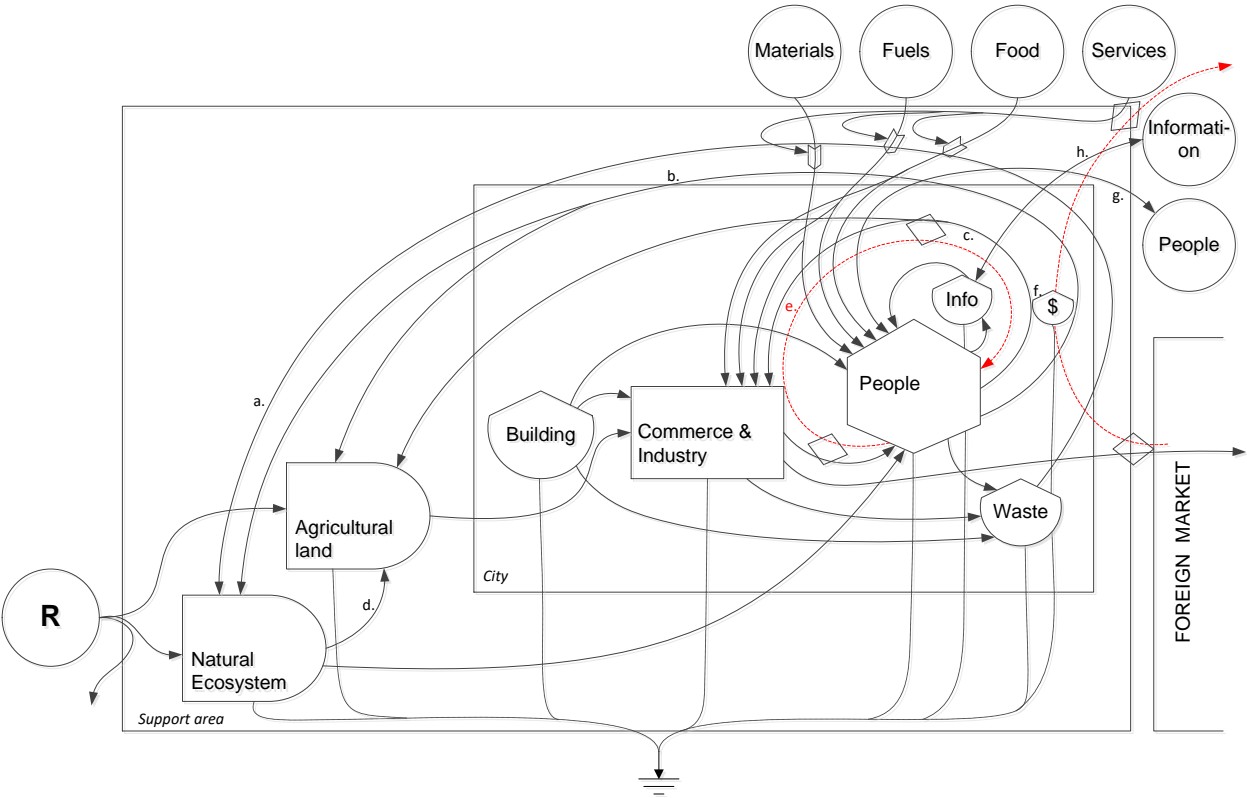

**Figure 4.** Emergy diagram of an urban ecosystem for Concórdia (Santa Catarina, Brazil). Source: adapted from Su et al. [37]. Note: R represents the local renewable resources. [a] Vegetable biomass providing ecosystem services for dissolution of negative externalities. [b] Landscape aesthetic contemplation. [c] Agricultural workforce. [d] Ecosystem services supporting agricultural production. [e] Monetary flow used to pay for goods and services in the support area. [f] Economic assets used to pay for services from outside of the support area. [g] People exchange. [h] Information exchange (i.e., social media, the specialized literature, social interaction, etc.).

### 3.2.3. Estimating a Holistic View-Based Payment for the Broiler Smallholder through Their Services

The model used to estimate a holistic view-based payment for the smallholder broiler family using their services was based on the emergy exchange ratio (EER) indicator proposed by Odum [25]. For Odum, this indicator can help to develop equity in trade, employing shared information among the agents and increasing the benefits for them. This indicator considers the emergy exchanged in a trade or purchase (the given emergy in product form and the received emergy in money form) and is generally expressed concerning one trading partner or the other trading partners. This is a measure of the relative trade advantage of one partner over the other [38]. Thus, the best result for the EER is 1.0, indicating a balanced trade under emergy units among the trading partners. From a holistic view, the EER can indicate an imbalanced emergy trade between the buyer (processing firm) and the smallholder broiler family using the buying power of money paid to this end.

To calculate the EER indicator, initially, the emergy accounting model was used to evaluate the emergy flows according to Nacimento et al. [39] (Figure 5). The model was filled using data previously recorded from the Embrapa Suínos e Aves reports [40,41]. These reports provide information regarding broiler production system features, buildings and management for the representative broiler systems in Santa Catarina. For the monetary data collection, interviews with producers, technical staff and experts in broiler production were conducted. The emergy flows were divided into the agroindustry (processing firm) and producer (smallholder broiler family) according to each input under their responsibilities,

as previously shown in the economic cost calculation. The detailed calculation procedure is available in Table S4 of Supplementary Material C.

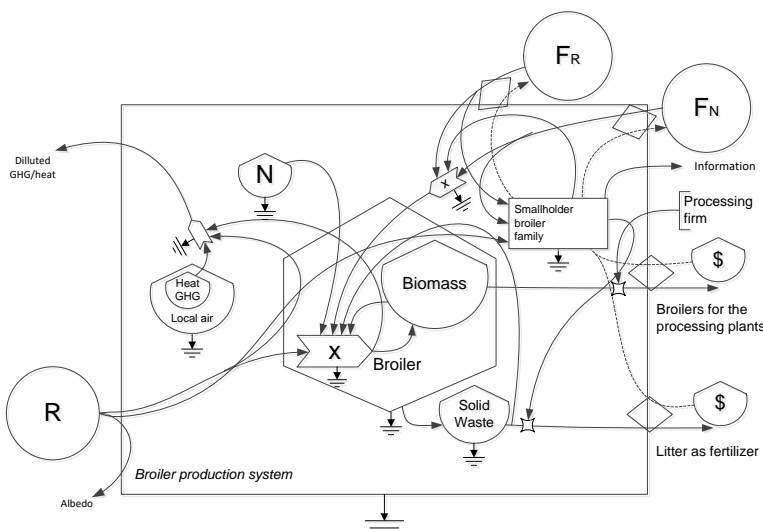

**Figure 5.** Emergy diagram for a representative broiler production system in Santa Catarina.

Second, the EER was used considering only the emergy flows of the broiler smallholder. Thus, the EER was the ratio between (1) the economic and socioenvironmental costs under the responsibility of the broiler smallholder in emergy terms and (2) the wealth provided in monetary terms and received by the smallholder broiler family for its services (broiler rising) is expressed as follows:

$$EER_p = \frac{Y_p}{Price \times EMR} \tag{1}$$

If $EER_p = 1$,

$$1 = \frac{Y_p}{a \times EMR} \tag{2}$$

in which $EER_p$ is the emergy exchange ratio for the smallholder (adimensional) and $Y_p$ is the emergy of the inputs under the responsibility of the smallholder (sej/yr). To this end, as the inputs were considered, those inputs under the smallholder broiler's responsibility were used to calculate the economic costs. *Price* is the price received as payment for the smallholder broiler family services in 2018 (USD/yr), *a* is the searched value of the more balanced payment as a monetary equivalent under the balanced condition $EER_p = 1$, and *EMR* is the emergy/money ratio for Santa Catarina previously assessed by Demétrio [33]. An average EMR (sej/$) can be calculated by dividing the total emergy use of a state or nation by its gross economic product. The EMR can be defined as the emergy supporting the generation of one unit of economic product (expressed in terms of currency) and is used as the economic equivalent of emergy. The contribution to a process represented by monetary payments is the emergy that people purchase with the money since that money is not paid back to the environment. Also, the amount of resources that money buys depends on the amount of emergy supporting the economy and the amount of money in circulation [42,43]. For Brown and Ulgiati [43], the EMR is useful for evaluating service inputs given in money units, where an average wage rate is appropriate.

Thus, emergy with and without cultural information was used to suggest a holistic view-based payment for the smallholder broiler family. For the model implementation and testing, the Solver tool (MS-Excel) by means of the non-linear GRG solution method was used. For further details, see Supplementary Material C.

## 4. Results and Discussion

### 4.1. Valuing Cultural Information

The Tr for the culture information storage of citizens of Santa Catarina was $2.77 \times 10^{10}$ sej/J (Table 1). As expected, this result is near the native culture storage presented by Odum [25] and depicted in graphical form by Abel [17]. It is also close to the value of traditional farming culture storage for the Oak Openings region found by Higgins [44].

**Table 1.** Emergy synthesis of culture information of Santa Catarina, Brazil citizens.

| | Item | Unit | Energy | Transformity | Emergy | Emdollar |
|---|---|---|---|---|---|---|
| | | | (units/yr) | (sej/unit) | (sej/yr) | in millions |
| | | | | | Annual Flow | |
| 1 | Renewable resource | J | | | $5.60 \times 10^{22}$ | 23,769.10 |
| 2 | Human metabolism | J | $2.71 \times 10^{16}$ | $2.06 \times 10^{6}$ | $5.60 \times 10^{22}$ | 23,769.10 |
| 3 | Information flow | J | $2.71 \times 10^{15}$ | $2.06 \times 10^{7}$ | $5.60 \times 10^{22}$ | 23,769.10 |
| | | | | | Steady-state Storage | |
| | | | (sej) | (sej/J) | (sej) | (emdollar) in millions |
| 4 | Population | J | $2.02 \times 10^{15}$ | $9.13 \times 10^{8}$ | $1.85 \times 10^{24}$ | 784,380.31 |
| 5 | Culture information | J | $2.02 \times 10^{14}$ | $2.77 \times 10^{10}$ | $5.60 \times 10^{24}$ | 2,376,910.02 |

Note: Number of citizens of Santa Catarina = 7.1 million people; emergy/money ratio from Santa Catarina state = $2.36 \times 10^{12}$ sej/USD [33]. The method proposed to estimate the culture information followed the equations suggested by Odum [25] and Odum and Doherty [32]. Meanwhile, sej = solar emjoule; J = joules. Transformity is the ratio obtained by dividing the total emergy that was used in a process by the energy yielded by the process [38]. The emdollar is a measure of the money that circulates in an economy as the result of some process and is obtained by multiplying an emergy flow or storage by the ratio of the total emergy to the gross domestic product for the national economy [38]. For details, see Supplementary Material B, namely the sheet marked "Cultural information". [1] The emergy of renewable resources for Santa Catarina was obtained from data previously published by Demétrio [33]. [2] Human metabolism. Used energy (J) = (7.1 × 10⁶ people) (2500 kcal/day) (4186 J/kcal) (365 day/y). Used energy (J) = $2.71 \times 10^{16}$ J/y. Transformity = (emergy of renewable resource)/(human metabolism). [3] Information flow: used energy (J) = (used energy) (10%). Used Energy (J) = $2.71 \times 10^{15}$ J/y. Transformity = (emergy of renewable resource)/(information flow). [4] Population: used energy (J) = (0.2 dry) (454 g/lb) (7.4 × 10⁴ people) (150 lb each) (5 kcal/g) (4186 J/kcal). Used energy (J) = $2.02 \times 10^{15}$ J. EMERGY = (average age) (renewable resources). EMERGY = (33 y) (renewable resources). EMERGY = $1.85 \times 10^{24}$ sej. Transformity = (emergy of population)/(used energy). [5] Culture information: used energy (J) = (energy used) (10%). Used energy (J) = $2.02 \times 10^{14}$ J EMERGY = (renewable resources) (100 yr). EMERGY = $5.60 \times 10^{24}$ sej. Transformity = (emergy of culture information)/(used energy).

An emergy-based culture evaluation can be made using the energy matter inputs and time required for cultural development [25]. For this, this study adopted a time window of 100 years. For the authors, the events and interactions as well as the information and knowledge exchange that occurred in this period between the earliest and newest people (i.e., immigrants and indigenous people) allowed the broiler production chain in southern Brazil to become stronger. For Abel [18], using the "areal" and "populational" calculation facilitated an estimation of the quality of the information by means of the Tr for an original production of information. Thus, the Tr represents the quality of the generational information about broiler production over the past 100 years to which the smallholder family had access. In addition, the Tr of cultural information is similar to the spectrum of the legacy and public status. Since the Tr represents the quality information, the information is the legacy built from the successes and failures of the first Brazilian broiler producers.

### 4.2. Emergy Synthesis of Broiler Smallholders

According to the results, cultural information showed the highest contribution for the emergy of the smallholder (62.9%), followed by wage (20.6%) and educational information (13.4%; Table 2).

**Table 2.** Emergy for broiler smallholder in Concórdia, Santa Catarina, Brazil.

| Item | Unit | Data | Transformity | | Emergy | Emdollar | % |
|---|---|---|---|---|---|---|---|
| | | (units/y) | (sej/unit) | | $(10^{13}$ sej/y) | (sej/USD.y) | |
| ENVIRONMENT RESOURCES (considering kinetic wind and Earth cycle emergy contributions in terms of emergy per person) | | | | ** | 0.37 | 1.55 | 0.0 |
| RESOURCES CONSUMPTION | | | | | 19,828.42 | 84,161.36 | 78.2 |
| Food | J | $3.06 \times 10^9$ | $1.52 \times 10^5$ | a | 46.45 | 197.15 | 0.2 |
| Electric power | J | $1.12 \times 10^7$ | $6.45 \times 10^4$ | b | 0.07 | 0.31 | 0.0 |
| Fuel | J | $2.92 \times 10^{10}$ | $1.41 \times 10^5$ | c | 411.73 | 1747.58 | 1.6 |
| Natural gas | J | $9.66 \times 10^7$ | $2.90 \times 10^4$ | b | 0.28 | 1.19 | 0.0 |
| Ethanol | J | $2.12 \times 10^8$ | $1.41 \times 10^5$ | c | 2.99 | 12.70 | 0.0 |
| Information | | | | | | | |
| Cultural information | J | $5.77 \times 10^6$ | $2.77 \times 10^{10}$ | Table 1 | 15,961.45 | 67,748.08 | 62.9 |
| Educational information | J | $9.84 \times 10^7$ | $3.46 \times 10^8$ | b | 3405.45 | 14,454.36 | 13.4 |
| MONETARY FLOW | | | | | | | |
| Wage | USD | $2.22 \times 10^4$ | $2.36 \times 10^{12}$ | d | 5228.87 | 22,193.86 | 20.6 |
| PRODUCED WASTE | | | | | | | |
| Solid waste treatment | J | $1.30 \times 10^9$ | $2.29 \times 10^6$ | e | 298.31 | 1266.19 | 1.2 |
| OUTPUTS * | | | | | | | |
| Product | | | | | | | |
| without Ci | | | | | | | |
| Work | J | $1.47 \times 10^9$ | $6.40 \times 10^7$ | | 9394.52 | 39,874.88 | 100 |
| Coproducts | | | | | | | |
| Sleep | J | $5.50 \times 10^8$ | $1.71 \times 10^8$ | | 9394.52 | | |
| Leisure | J | $3.82 \times 10^8$ | $2.46 \times 10^8$ | | 9394.52 | | |
| with Ci | | | | | | | |
| Work | J | $1.47 \times 10^9$ | $1.73 \times 10^8$ | | 25,355.97 | 107,622.96 | 100 |
| Coproducts | | | | | | | |
| Sleep | J | $5.50 \times 10^8$ | $4.61 \times 10^8$ | | 25,355.97 | | |
| Leisure | J | $3.82 \times 10^8$ | $6.64 \times 10^8$ | | 25,355.97 | | |

Note: Number of citizens of Concórdia, Brazil = 74,106 people; emergy/money ratio from Santa Catarina = $2.36 \times 10^{12}$ sej/USD [33]; BRL-to-USD exchange ratio = 3.65:1.00. * Ci = cultural information; for details see Supplementary Material B, namely the sheet titled "Manpower". ** To avoid double-accounting, the environment resources were considered as the summation of wind, kinetic energy [45] and Earth cycles [46]. [a] Brandt-Williams [28]. [b] Giannetti et al. [45]. [c] Odum [25]. [d] Demétrio [33]. [e] Huang et al. [47]. J = joules, and transformity is the ratio obtained by dividing the total emergy that was used in a process by the energy yielded by the process [38]. The emdollar is a measure of the money that circulates in an economy as the result of some process and is obtained by multiplying an emergy flow or storage by the ratio of the total emergy to the gross domestic product for the national economy [38].

Information comprised 76% of the smallholder emergy as a whole. The Tr for the work of broiler smallholders with cultural information was $1.73 \times 10^8$ sej/J. Based on Giannetti et al. [45], this value for the Tr is similar to the spectrum of people's education that considers cultural information, and the quality of the human service was similar to that of the post-college educated students [48,49]. This leads to the following question: Is the human service provided by smallholders in broiler production similar to that of a technician or a college graduate in terms of quality? Although the broiler smallholder farmer could not have a formal education at any post-graduate education level, it is possible that the longer period used to learn about production techniques with parents or technical assistants may have come close the quality of information exchanged between professionals and broiler smallholders. For Odum [25], knowledge is the collection of information transmitted and the emergy flows that support the person. Thus, the raised supposition could be understood better when considering the broiler smallholder's age and his or her service years. In this sense, quality information (or knowledge) is considered as an interaction based on the age and service time.

There was a higher emergy value for labor ($2.5 \times 10^{17}$ sej/J) when compared with the focus on agricultural system perspectives [28,50,51]. According to Odum [25], human

service is evaluated (1) by multiplying the energy expended by a human being by the Tr of that person's education and experience or (2) by dividing the total national emergy flow by the number of people and the metabolism. The higher emergy observed in this study was expected since it included the emergy for cultural and educational information, which is generally not considered in other studies. In addition, leisure and sleep were included in this study [28], proposing a holistic view regarding the costs for human services and considering the "real" wealth. In this sense, it is important to include inputs that support the social welfare and quality of life. Also, this holistic perspective may contribute to further studies in agricultural systems that aim at a deeper comprehension of the impact of human services on system sustainability.

### 4.3. The Impact of Cultural Information on Emergy Indicators for Broiler Production

In this study, the emergy contribution of each input was allocated to the processing firm (agroindustry) and smallholder according to the allocation approach shown in Figure 2. This allowed us to (1) estimate a payment for the broiler smallholders for their services by means of a holistic view and (2) estimate the impact of the cultural information from a smallholder in production as a whole. In this section, the discussion is focused on the emergy indicators.

Table 3 indicates that the emergy indicators were not impacted when the cultural information from the broiler smallholder was considered. In other words, including the cultural information was not sufficient to promote the better use of local renewable resources (EYR = 1.22), minimize the environmental load (ELR = 4.65) or even increase sustainability (ESI = 0.26) in the broiler production system.

**Table 3.** Comparison of emergy indicators considering the cultural information (Ci) of smallholder in a broiler production system.

| Index | Equation | With Ci | | | Without Ci | | | With Ci Being Renewable [†] | | |
|---|---|---|---|---|---|---|---|---|---|---|
| | | Total | AGi | SH | Total | AGi | SH | Total | AGi | SH |
| Y | $R + N + F$ | 12.90 | 9.22 | 3.66 | 11.30 | 9.22 | 2.07 | 12.89 | 9.23 | 3.66 |
| Broiler Tr | $\frac{Y}{Ep}$ | 4.72 | 3.38 | 1.34 | 4.14 | 3.38 | 0.76 | 4.73 | 3.38 | 1.34 |
| R% | $\frac{R}{R+N+F}$ | 18% | 22% | 6% | 20% | 22% | 11% | 33% | 22% | 59% |
| N% | $\frac{N}{R+N+F}$ | 0% | 0% | 0% | 0 | 0% | 0% | 0% | 0% | 0% |
| F% | $\frac{F}{R+N+F}$ | 82% | 78% | 94% | 80% | 78% | 89% | 67% | 78% | 41% |
| ELR | $\frac{N+F_N}{R+F_R}$ | 4.65 | 3.49 | 15.01 | 3.95 | 3.49 | 8.04 | 2.06 | 3.49 | 0.69 |
| EIR | $\frac{F}{R+N}$ | 5497 | - | 1562 | 4816 | - | 881 | 5500 | - | 1562 |
| EYR | $\frac{Y}{F_N}$ | 1.22 | 1.29 | 1.07 | 1.25 | 1.29 | 1.12 | 1.49 | 1.29 | 2.45 |
| ESI | $\frac{EYR}{ELR}$ | 0.26 | 0.37 | 0.07 | 0.32 | 0.37 | 0.14 | 0.72 | 0.37 | 3.53 |
| EER | $\frac{Y}{\$ \times sej/\$}$ | 2.08 | 1.49 | 0.59 | 1.82 | 1.49 | 0.33 | 2.08 | 1.49 | 0.59 |
| EER$_p$ | $\frac{Y_{smallholder}}{\$/bird \times sej/\$}$ | | | | 1.00 | | | 1.00 | | |
| Emprice | if $EER = 1$ | | | | 1.62 | | | 0.91 | | 1.62 |

Note: AGi = agroindustry (processing firm) result indexes; SH = smallholder result indexes. [†] The emergy flow of Ci was considered a renewable resource; Y is the emergy ($\times 10^{17}$ sej/y); Tr is the transformity ($\times 10^5$ sej/J); R% is the renewable resources (%); N% is the nonrenewable resources (%); F% is the purchased resources (%); ELR is the environmental loading ratio (sej/sej); EIR is the emergy investment ratio (sej/sej); EYR is the emergy yield ratio (sej/sej); ESI is the emergy sustainability index (sej/sej); EER is the emergy exchange ratio (sej/sej); EER$_p$ is the emergy exchange ratio for the smallholder (sej/sej); R is the local renewable resouces (sej); N is the local non-renewable resources (sej); F is the purchased input (sej); F$_R$ is the renewable fraction from purchased inputs (sej); F$_N$ is the non-renewable fraction from purchased inputs (sej); Ep is the produced energy (J); sej stands for solar emjoules; and J stands for joules. Estimated broiler payment is expressed in USD/bird if EER = 1. The emprice is a measure of the money that circulates in an economy as the result of some processes [38]. The emprice was obtained dividing the broiler system emergy under the smallholder's responsibility by the emdollars received as payment for their service. The dollar value was used for the real price and the estimated economic price.

Services (38%), feed (36%) and labor (20%) contributed most significantly to the emergy of the broiler production system. In this sense, higher accuracy in calculating these resource inputs is welcome to obtain results with higher accuracy for the emergy indicators (Table 4).

**Table 4.** Emergy for a representative broiler system in Santa Catarina, Brazil.

| Item | Unit | Data | Transformity | | Emergy | Emdollar | % | %Ren |
|---|---|---|---|---|---|---|---|---|
| | | (units/yr) | (sej/unit) | | ($10^{13}$ sej/yr) | (sej/USD.yr) | | |
| RENEWABLE RESOURCES | | | | | 15.50 | 65.77 | 0.01 | |
| Sun | J | $1.40 \times 10^6$ | 1 | a | 0.00 | 0.00 | 0.00 | R |
| Rain, geopotential energy | J | $4.42 \times 10^6$ | $1.30 \times 10^4$ | b | 0.01 | 0.02 | 0.00 | R |
| Rain, chemical potential | J | $6.63 \times 10^6$ | $9.71 \times 10^3$ | b | 0.01 | 0.03 | 0.00 | R |
| Wind, kinetic energy ** | J | $1.13 \times 10^9$ | $1.28 \times 10^3$ | b | 0.14 | 0.61 | 0.00 | R |
| Forced cooling ** | J | $1.20 \times 10^{11}$ | $1.28 \times 10^3$ | b | 15.35 | 65.16 | 0.01 | R |
| NONRENEWABLE STORAGES | | | | | 7.94 | 33.71 | 0.01 | |
| Groundwater | J | $9.22 \times 10^5$ | $9.74 \times 10^4$ | c | 0.01 | 0.04 | 0.00 | N |
| Soil occupation | J | $6.10 \times 10^8$ | $1.30 \times 10^5$ | b | 7.93 | 33.68 | 0.01 | N |
| Topsoil losses | J | 0.00 | $1.30 \times 10^5$ | b | 0.00 | 0.00 | 0.00 | N |
| Sum of free inputs (wdc) | | | | | 23.44 | 99.49 | 0.02 | |
| PURCHASED RESOURCES | | | | | 128,916.13 | 388,297.57 | 99.98 | |
| Buildings | | | | | 565.37 | 2399.72 | 0.44 | F |
| Wood | g | $9.60 \times 10^5$ | $6.69 \times 10^8$ | d | 64.21 | 272.54 | 0.05 | F |
| Steel | g | $7.08 \times 10^4$ | $2.39 \times 10^9$ | d | 16.90 | 71.71 | 0.01 | F |
| Sand | g | $3.84 \times 10^5$ | $8.51 \times 10^8$ | d | 32.69 | 138.74 | 0.03 | F |
| Cement | g | $1.18 \times 10^5$ | $1.57 \times 10^9$ | d | 18.55 | 78.72 | 0.01 | F |
| Gravel | g | $3.54 \times 10^5$ | $1.28 \times 10^9$ | e | 45.20 | 191.85 | 0.04 | F |
| Block | g | $8.25 \times 10^4$ | $1.76 \times 10^9$ | d | 14.55 | 61.74 | 0.01 | F |
| Tile | g | $1.55 \times 10^6$ | $2.33 \times 10^9$ | d | 359.50 | 1525.89 | 0.28 | F |
| Depreciation | g | | | | 13.79 | 58.53 | 0.01 | F |
| Labor | | | | | 25,356.75 | 107,626.26 | 19.67 | |
| Technical assistance | J | $1.48 \times 10^8$ | $3.12 \times 10^4$ | f | 0.46 | 1.96 | 0.00 | 5%R |
| Harvesting services | J | $1.01 \times 10^8$ | $3.12 \times 10^4$ | f | 0.32 | 1.34 | 0.00 | 5%R |
| Farmer | J | $1.47 \times 10^9$ | $1.73 \times 10^8$ | Table 2 | 25,355.97 | 107,622.96 | 19.67 | F |
| Registered manpower | J | 0.00 | $3.12 \times 10^4$ | f | 0.00 | 0.00 | 0.00 | 5%R |
| Unregistered manpower | J | 0.00 | $3.12 \times 10^4$ | f | 0.00 | 0.00 | 0.00 | 5%R |
| Wood | J | $7.50 \times 10^4$ | $2.64 \times 10^4$ | b | 0.00 | 0.00 | 0.00 | F |
| Natural gas | J | $9.71 \times 10^5$ | $2.90 \times 10^4$ | b | 0.00 | 0.01 | 0.00 | F |
| Wooden shave (litter) | J | $3.50 \times 10^{11}$ | 3.80 | g | 1329.81 | 5644.35 | 1.03 | 43%R |
| Fuel | J | 0.00 | $1.41 \times 10^5$ | a | 0.00 | 0.00 | 0.00 | F |
| Hydroelectric power | J | $4.00 \times 10^{10}$ | $6.45 \times 10^4$ | b | 257.69 | 1093.77 | 0.20 | 65%R |
| Pesticides and vaccines | g | 0.00 | $1.12 \times 10^{10}$ | h | 0.00 | 0.00 | 0.00 | F |
| Chicks | g | $9.70 \times 10^{10}$ | $4.64 \times 10^5$ | i | 4502.42 | 19110.44 | 3.49 | 16%R |
| Feed | | | | | 46,791.08 | 198,603.92 | 36.29 | |
| Corn | J | $4.48 \times 10^{12}$ | $5.10 \times 10^4$ | b | 22,867.49 | 97,060.65 | 17.74 | 22%R |
| Soybean meal | J | $2.60 \times 10^{12}$ | $9.20 \times 10^4$ | b | 23,923.59 | 101,543.27 | 18.55 | 33%R |
| Mechanical equipment | g | $4.55 \times 10^5$ | $1.82 \times 10^9$ | j | 82.98 | 352.20 | 0.06 | F |
| Depreciation | | | | | 6.91 | 29.35 | 0.01 | F |

**Table 4.** *Cont.*

| Item | Unit | Data | Transformity | | Emergy | Emdollar | % | %Ren |
|---|---|---|---|---|---|---|---|---|
| Transport | | | | | 1034.34 | 4390.25 | 0.80 | |
| Mechanical equipment | g | $1.24 \times 10^6$ | $5.09 \times 10^9$ | k | 629.01 | 2669.83 | 0.49 | F |
| Labor | J | $1.31 \times 10^8$ | $3.12 \times 10^4$ | f | 0.41 | 1.74 | 0.00 | 5%R |
| Fuel | J | $2.87 \times 10^{10}$ | $1.41 \times 10^5$ | a | 404.92 | 1718.68 | 0.31 | F |
| Services | $ | $2.08 \times 10^5$ | $2.36 \times 10^{12}$ | f | 48,988.76 | 48,988.76 | 37.99 | 17%R |
| OUTPUTS * | | | | | | | | |
| without Ci | | | | | | | | |
| Total Yield, dry weight (broiler) | J | $2.73 \times 10^{12}$ | $2.35 \times 10^5$ | | | | | |
| Total Yield, dry weight (broiler) | kg | $2.55 \times 10^5$ | $2.51 \times 10^{12}$ | | | | | |
| Total Yield, dry weight (litter as fertilizer) | J | $3.50 \times 10^{11}$ | $1.83 \times 10^6$ | | | | | |
| Total Yield, dry weight (litter as fertilizer) | kg | $2.09 \times 10^4$ | $3.06 \times 10^{13}$ | | | | | |
| with Ci | | | | | | | | |
| Total Yield, dry weight (broiler) | J | $2.73 \times 10^{12}$ | $2.93 \times 10^5$ | | | | | |
| Total Yield, dry weight (broiler) | kg | $2.55 \times 10^5$ | $3.13 \times 10^{12}$ | | | | | |
| Total Yield, dry weight (litter as fertilizer) | J | $3.50 \times 10^{11}$ | $2.28 \times 10^6$ | | | | | |
| Total Yield, dry weight (litter as fertilizer) | kg | $2.09 \times 10^4$ | $3.83 \times 10^{13}$ | | | | | |

Note: Number of citizens of Concórdia, Brazil = 74,106 people; emergy/money ratio from Santa Catarina state = $2.36 \times 10^{12}$ sej/USD [33]; BRL-to-USD exchange ratio = 3.65:1.00. * Ci = cultural information; for details, see Supplementary Materials C. ** To avoid double-accounting, the environment resources were considered as the summation of wind, kinetic energy and forced cooling. [a] By definition from Odum [25]. [b] Giannetti et al. [45]. [c] Buenfil [52]. [d] Brown and Buranakarn [53]. [e] Pulselli et al. [54]. [f] Demétrio [33]. [g] Comar and Komori [55]. [h] Brandt-Williams [28]. [i] Castellini et al. [56]. [j] Bargigli and Ulgiati [57]. [k] Brown [28]. %Ren is the percentage of renewable emergy. This is the ratio of renewable emergy to total emergy use. The %Ren is shown for each of the following inputs: wooden shavings (litter) [55]; hydroelectric power [58]; corn [56]; soybean meal [59], labor and services [60] and chicks [39]. J stands for joules. Transformity is the ratio obtained by dividing the total emergy that was used in a process by the energy yielded by the process [38]. The emdollar is a measure of the money that circulates in an economy as the result of some process and is obtained multiplying the emergy flow or storage by the ratio of the total emergy to the gross domestic product for the national economy [38].

For this study, the partial renewabilities of services (in currency) and feed ingredients were considered. Although the emergy from the renewable resource in the transformity labor assessment was included, its partial renewability was near zero. This can be explained by the fact that the partial renewability of the F inputs that composed its emergy-based assessment was not considered. Thus, hypothetically and empirically, if the emergy contribution of the information (culture and educational information (76%)) is considered a renewable resource, then the emergy sustainability index can increase by 2.7 times (ESI of 0.72 vs. 0.26). However, it is still not clear to what extent generational information is renewable. According to Abel [17], "culture" is a "kind" of information that requires a population to maintain it within continuous information cycles of selection and renewal. These cycles could increase knowledge durability, creating a cultural model in a population of individuals over time. Thus, the original information is not lost but replaced by the most efficient information. The first one is needed to give rise to the last one, which solves the problem more efficiently. This could explain the importance of generational information as "useful information" for broiler production optimization over time. For Abel [17], useful information is a product of the self-organization of systems, wherein its function is to remember successful configurations. Agriculture, manufacturing, education and all other cultural "industries" could not exist without humans. Thus, the pathway to the (re)construction of more sustainable and socially fair agriculture is through greater attention to indigenous knowledge systems [20].

From an overall perspective, the results show that human services are central to the functioning of any human-influenced process. Without human control, there is no application of information, and there is no organization of material and energy inputs [61]. Also, system studies that do not include human service inputs do not properly depict the system under study. In addition, there is a risk that omission of labor inputs in environmental assessment promotes a leakage of environmental effects linked to the human labor needed [61] and impairs sustainability assessment. Thus, the inclusion of labor contributions in emergy-based assessments for agricultural systems in a more accurate perspective must be estimated. "Areal" and "populational" aspects as well as culture and microeconomics must be considered in these models.

### 4.4. The (Im)balance between the Estimated Economics-Based and Environment-Based Payments

The results highlight an imbalance in the monetary exchange between the processing firm and broiler smallholder for both the economic and emergy approaches. Both the estimated economics-based and environment-based payments per broiler were higher than the practiced payment value (Figure 6).

The values shown in Figure 6 present a picture that can be interpreted in three stages. The first concerns the difference between the real and estimated prices (including infrastructure depreciation). The estimated payments per broiler surpass the current payment value in practice. The amount paid to the producer does not cover some tangible costs; the producer only receives payment for handling the product, having to bear the costs of maintaining the infrastructure and payments for the services necessary to serve the industry. In this context, the decapitalization of the broiler smallholder is in serious danger. Given this situation, it is suggested to revise the existing commercial agreement between the parties involved.

In the second stage, the broiler price is estimated by the emergy synthesis. The emprice (without cultural information) includes natural resources (sunlight, rain and wind) and is calculated by considering not only the quantity of resources employed but also their quality. This estimate, with a monetary equivalent of 0.5 EmUSD/broiler, measures the emergy that supports the poultry production process in the given economy. The difference between the estimated price and the emprice (USD 0.25/broiler) could provide a measure for eco-compensation for the industrial activity [62] or be reserved and eventually employed as a reinforcement of the productive basis (natural and man-made) that would subsidize the negative externality reduction provided by the inputs.

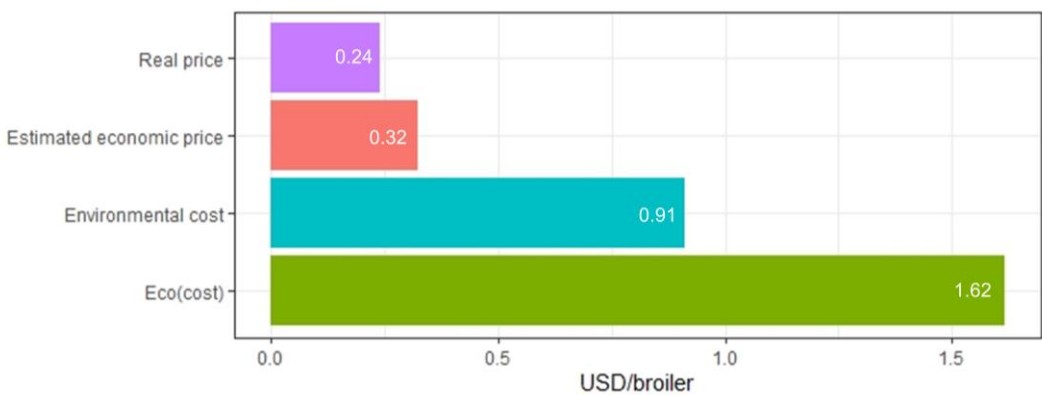

**Figure 6.** Comparison between the payments received by the broiler smallholders for their services (real price) and the estimated payment considering the economic cost (plus 10% of profit; estimated economic price), the estimated payment considering the emprice without cultural information ("environmental cost") and the estimated payment considering the emprice with cultural information ("eco(cost)"). The emprice is a measure of the money that circulates in an economy as the result of some processes [38]. The emprice was obtained by dividing the broiler system emergy under a smallholder's responsibility by the emdollars received as payment for services. The dollar value was used for the real price and the estimated economic price. BRL-to-USD exchange rate = 3.65:1.00.

The discussion becomes more complex when cultural information is included in the third interpretation stage. Culture is perceived as socially spread, differently internalized and actively debated or bargained among various groups and subgroups within a community [17]. It is a constructed outcome of interactions with others and the cultural knowledge generated on different levels and over time. The fact that the region is culturally dedicated to poultry production carries with it the idea that the workers in this region are culturally indoctrinated to believe that this is their vocation and that the activity is a family business. This cultural indoctrination, passed down through generations, naturally creates specialized producers dedicated to this activity who understand and accept the working conditions, which would be difficult to implement in other regions without significant efforts. As an example, the Brazilian midwest region has several competitive advantages that could promote broiler production (e.g., low feed costs, available area for production and governmental tax benefits). However, although some microregions have been able to take advantage of those benefits for promoting broiler production, they are concentrated in the southern region [63,64]. Thus, ignoring the cultural aspects of the population and its contribution as one of the most important variables for strengthening broiler production could be unfair. In addition, the intangible value of this effort can be estimated when including the value of cultural information in the emprice of the final product. The emprice is a monetary equivalent to the process, enabling humans to not only build innovation upon "cumulative cultural evolution" or "social learning" [17] but also a guarantee for the industry to receive culturally moulded and trained partners for this specific activity. The discrepancy of 0.70 EmUSD/broiler could serve multiple purposes, not only being applied to educational programs, training and health initiatives but also a reserve, fund or reinforcement that could be used to strengthen the productive foundation, encompassing both natural and man-made aspects.

From an overall perspective, a methodological approach based on the emergy theory indicates the existing imbalance in monetary exchange between the processing firm and the broiler producer family. The current business model places the responsibility for tangible costs (such as infrastructure maintenance and services) on poultry-producing families, whereas the industry benefits from the intangible costs (i.e., the accumulated cultural knowledge) without recognizing it or compensating for it. Introducing a new business and contract model based on the procedure outlined in this paper could push discussions on a more equitable balance between the integrator and the integrated parties. Thus, this model

would consider more appropriately the importance of the generational acknowledgement of broiler production success and then accurately remunerate the contributions of all stakeholders involved.

### 4.5. General Insights into Quantifying the Value

There are several objective forms of evaluation that aim to assign a quantifiable and measurable value to a given product, service or asset, eliminating or minimizing the subjectivity involved, which include the following:

1. Market valuation: This form of valuation involves the analysis of prices and recent transactions for similar products or assets on the market. The idea is to use objective data from real transactions to establish a value based on market conditions.
2. Valuation by cost: This method aims to determine value based on production costs, including materials, labor, depreciation, operating expenses and other expenses. In the industry, the cost of production is a good indicator of the value of the product.
3. The valuation of environmental costs based on emergy (without services or information) uses exclusively a physical inventory of the resources used in its accounting. This type of valuation takes into account the energy embodied in natural resources and considers their quantity and quality. Emergy makes it possible to assess the environmental costs associated with production. The emergy valuation approach (without services and information) can be useful, especially when the focus is on analyzing the physical resources, production technologies and geographical aspects (such as rainfall and land use) involved in a given production system.
4. The valuation of the environmental cost in terms of emergy with services (including the money paid in environmental accounting) offers a more comprehensive view and may be more appropriate when it is desired to understand the social, economic and environmental impacts of the organization in a more complete and integrated way.
5. Evaluating the "(eco)cost" in emergy by including information can take into account the scientific knowledge, technology, intellectual capital and culture, which are essential to the operation and functioning of production systems. Including information in the estimation of emergy aims to reflect the contribution of these elements and allows for a more comprehensive assessment of the efficiency and sustainability of the production system. This approach is highly relevant in emergy-based environmental accounting as it recognizes that production depends not only on tangible physical resources but also on the knowledge, skills and information available. Therefore, emergy-based environmental accounting that includes information seeks to understand and quantify both the physical and intangible aspects involved in the production and functioning of economic and ecological systems.

The "iceberg of value (or pyramid)" thinking becomes interesting and useful for understanding the functioning of production systems and their underlying bases. This approach recognizes that there are different layers of value and resources involved in any economic system and that not all of these resources are easily observable or quantifiable on the surface. The "iceberg of value" thinking suggests that there are hidden costs in invisible layers of resources and contributions that support the production system beyond the market and cost values. These layers can include intellectual capital, scientific knowledge, innovation, technology, human skills and the natural resources used. By recognizing the importance of underlying resources that are not directly visible, the value pyramid concept highlights the relevance of sustainability and the efficient management of these resources for the long-term success of an economic system. Ignoring or underestimating these invisible layers can lead to problems such as overexploitation of resources or a lack of innovation and competitiveness. In turn, the "iceberg of value" thinking encourages a more holistic and integrated analysis of production systems, considering not only the direct economic aspects but also the social and environmental impacts associated with these invisible layers of value. This can lead to a better understanding of the challenges and opportunities of a production system in relation to its sustainability and resilience.

The "iceberg of value" thinking offers an interesting way of visualizing the different levels of value and resources involved in production systems. Recognizing the importance of invisible resources can lead to a more balanced and sustainable approach to economic development and resource management. As with any conceptual model, its practical application may vary depending on the specific context and the objectives of the analysis, but it can be a valuable tool for a deeper understanding of economic systems and their foundations.

## 5. Conclusions

Human beings use information to allocate the available resources in the best way to make the human-influenced process as efficient as possible. Therefore, even in the more technological perspectives, broiler production will not give up the human factor for its continuity. However, the contribution of knowledge and cultural information from people has not been fully evaluated. It would be unfair to disregard the cultural aspects of the producers and their contributions as some of the most important resources for strengthening the agricultural production chain. Methodological approaches that consider intangible or hidden costs (natural resources and sociocultural aspects), such as emergy-based assessment, could help to quantitatively indications of the (im)balance in monetary exchange between the processing firm and the broiler smallholder.

This study highlights that cultural information is the most significant contributor to the emergy of broiler smallholders (~63%). In turn, cultural information increases the quality of the smallholder service to a spectrum of people education ($1.73 \times 10^8$ sej/J) similar to that of post-college-educated students. When considered, the large amount of time applied to learn about production techniques with ancient parents or technical assistants result in similar quality information between professionals and broiler smallholders.

Another important result is the obtained imbalance in the monetary exchange between the processing firm and broiler smallholder when considering both the economics- and emergy-based perspectives. The obtained values (USD 0.32/broiler and 1.62 EmUSD/broiler) were higher than the practiced payment value of USD 0.24/broiler. The numbers suggest that sustainability of the broiler production chain could be impaired in the long term due to decapitalization of the broiler smallholder and a reduction in the environmental (i.e., ecosystem services) and social (i.e., smallholder retention) support of the poultry production process. Decision makers must consider the promotion of public policies that subside initiatives to strengthen the productive foundation, including programs for social (e.g., educational programs, training, health initiatives and environmental welfare (i.e., landscape protection and rural tourism promotion)).

Although contributing to advancements in the important topic of accounting for cultural information in terms of quantifying value, this study still has limitations regarding the calculation procedures shown in Table 1. Even though advancements have been made ever since its first appearance in Odum's (1996) book [25], quantifying cultural information is still in its infancy and demands some assumptions regarding the more subjective aspects. Future efforts should cover such subjective aspects based on social sciences to bring about more accurate results. Additionally, it is suggested to carry out efforts in comparing the emergy indicators from different agricultural production systems under different scales, technological assets, and for different regions.

**Supplementary Materials:** The following supporting information can be downloaded at: https://www.mdpi.com/article/10.3390/su152215780/s1, Table S1. Economic cost calculation models [29]; Table S2. Emergy for storaged culture information of Santa Catarina citizen [25,33–35]; Table S3. Emergy for broiler farmer in Concórdia, Brazil [25,28,33–35,45–47,65–72]; Table S4. Emergy synthesis for broiler production in Santa Catarina State, Brazil [25,28,33,38,39,45,52–60,65,73–79].

**Author Contributions:** Conceptualization, B.F.G.; Methodology, Rafael Araujo Nacimento, M.D.C. and B.F.G.; Validation, R.A.N. and M.D.C.; Investigation, R.A.N. and M.D.C.; Writing—original draft, Rafael Araujo Nacimento, M.D.C., C.A., F.A., A.H.G. and B.F.G.; Writing—review & editing, A.H.G.

and B.F.G.; Supervision, B.F.G.; Project administration, M.D.C. and A.H.G. All authors have read and agreed to the published version of the manuscript.

**Funding:** This research received no external funding.

**Institutional Review Board Statement:** Not applicable.

**Informed Consent Statement:** Not applicable.

**Data Availability Statement:** The data presented in this study are available in the Supplementary Material.

**Conflicts of Interest:** The authors declare no conflict of interest.

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
