# Peer review of "Hidden Costs Associated with Smallholder Family-Based Broiler Production: Accounting for the Intangibles"

_sustainability, doi:10.3390/su152215780_

Round 1

Reviewer 1 Report

Comments and Suggestions for Authors

I have made a few suggestions to help the authors correct some errors and suggestions to help improve on the quality of the manuscript. In particular, authors are invited to check the figures' numbering and referencing in the text.

Comments on the Quality of English Language

The quality of the English Language is satisfactory and allows understanding of what the authors are communicating.

Author Response

General comments

First of all, the authors would like to thank you for your readness in helping us to improve the manuscript. To meet the suggestions of the reviewer, the text was reviewed as a whole. All the comments, suggestions, and observations helped us to improve the study as a whole. We sought to attain all of them expertly. Thus, we hope that the manuscript attains the desired level.

In addition, we inform you that the manuscript has been professionally proofread (please find the English-proof certificate at the bottom of this document).

Reviewer 1

I have made a few suggestions to help the authors correct some errors and suggestions to help improve on the quality of the manuscript. In particular, authors are invited to check the figures' numbering and referencing in the text.

Comments on the Quality of English Language:

The quality of the English Language is satisfactory and allows understanding of what the authors are communicating.

R: The authors agree with the suggestions. All the corrections were inserted.

Reviewer 2 Report

Comments and Suggestions for Authors

This seems to be a very complicated paper on a relatively simple idea about the importance of cultural traditions in maintaining the success of an area of broiler chicken production in southern Brazil.

The list of references seems fine, the same as the methodology; the tables and illustrations are excellent and thorough.

However, I feel that the use of the concept of 'Emergy' is confusing and the many detailed diagrams and tables complicate the message of the paper needlessly. The point of the importance of cultural background could be shown more simply.

Some problems with English, i.e. 'subside' in Abstract and elsewhere in manuscript when subsidy is meant.

Comments on the Quality of English Language

Some problems with English, i.e. line 35 'besides of generating jobs'. Also 'subside' instead of 'subsidy.'

Author Response

General comments

First of all, the authors would like to thank you for your readness in helping us to improve the manuscript. To meet the suggestions of the reviewer, the text was reviewed as a whole. All the comments, suggestions, and observations helped us to improve the study as a whole. We sought to attain all of them expertly. Thus, we hope that the manuscript attains the desired level.

In addition, we inform you that the manuscript has been professionally proofread (please find the English-proof certificate at the bottom of this document).

Reviewer 2

This seems to be a very complicated paper on a relatively simple idea about the importance of cultural traditions in maintaining the success of an area of broiler chicken production in southern Brazil.

The list of references seems fine, the same as the methodology; the tables and illustrations are excellent and thorough.

However, I feel that the use of the concept of 'Emergy' is confusing and the many detailed diagrams and tables complicate the message of the paper needlessly. The point of the importance of cultural background could be shown more simply.

Some problems with English, i.e. 'subside' in Abstract and elsewhere in manuscript when subsidy is meant.

Comments on the Quality of English Language:

Some problems with English, i.e. line 35 'besides of generating jobs'. Also 'subside' instead of 'subsidy.'

R: The authors agree with the observation. The manuscript was improved as a whole. Emergy concepts were inserted to become the message clearer.

Reviewer 3 Report

Comments and Suggestions for Authors

Overall opinion:

The topic is very exciting, especially for a journal like Sustainability. It deals with measuring the value of intangible production factors, among them the value of information, and especially cultural information. This is an important issue, and the author uses an interesting and well established model by Odum, for a Brazilian broiler production system. However, the structure and the presentation of the findings is not very good. The paper often refers back to earlier studies like the work of Odum (see references  23, 24, 22) or references of the work of Abel (see references 16 and 17). This is quite in order, but still the main components of the applied model should be explained in a brief form in the manuscript itself. At least the basic concept of the model should be summarised, together with the target variables, the inputs, and the key structure of the computations should be described in a concise form.  Instead, the reader is referred back to the mentioned references so that the present manuscript cannot be understood properly without reading the mentioned references. There is a nice picture about the general concept of the model in the introduction. However, in the Material and Method section the model is described without a proper listing of the terminology and without the basic logic of the computations explained. With this arrangement results are difficult to understand or interpret.

Please see details in the attached file.

Comments on the Quality of English Language

Language:

The English language is generally good. A few minor occurrences of strange usage were found. See details later. Details of language usage are included in the review later, but a few strange word forms are mentioned here as an example. ‘Estimative’ : is it ‘estimation of’, or ‘estimating’, or ‘estimate’? Another strange word is: ‘summatory’, which should rather be: ‘ The sum of’, or ‘summing up’, or some similar expression. See details in the attached review file.

Author Response

General comments

First of all, the authors would like to thank you for your readness in helping us to improve the manuscript. To meet the suggestions of the reviewer, the text was reviewed as a whole. All the comments, suggestions, and observations helped us to improve the study as a whole. We sought to attain all of them expertly. Thus, we hope that the manuscript attains the desired level.

In addition, we inform you that the manuscript has been professionally proofread (please find the English-proof certificate at the bottom of this document).

Reviewer 3

The topic is very exciting, especially for a journal like Sustainability. It deals with measuring the value of intangible production factors, among them the value of information, and especially cultural information. This is an important issue, and the author uses an interesting and well established model by Odum, for a Brazilian broiler production system. However, the structure and the presentation of the findings is not very good. The paper often refers back to earlier studies like the work of Odum (see references  23, 24, 22) or references of the work of Abel (see references 16 and 17). This is quite in order, but still the main components of the applied model should be explained in a brief form in the manuscript itself. At least the basic concept of the model should be summarised, together with the target variables, the inputs, and the key structure of the computations should be described in a concise form.  Instead, the reader is referred back to the mentioned references so that the present manuscript cannot be understood properly without reading the mentioned references. There is a nice picture about the general concept of the model in the introduction. However, in the Material and Method section the model is described without a proper listing of the terminology and without the basic logic of the computations explained. With this arrangement results are difficult to understand or interpret.

Please see details in the attached file.

Language: The English language is generally good. A few minor occurrences of strange usage were found. See details later. Details of language usage are included in the review later, but a few strange word forms are mentioned here as an example. ‘Estimative’: is it ‘estimation of’, or ‘estimating’, or ‘estimate’? Another strange word is: ‘summatory’, which should rather be: ‘ The sum of’, or ‘summing up’, or some similar expression. See details in the attached review file.

Introduction

After line 40, two pieces of information should be presented about the broiler section in Brazilian economy, namely what percentage of the GDP, and what percentage of the labour force does it represent? R: The authors agree with the suggestions. The paragraph “In 2022, the Brazilian poultry chain made available 4.1 million of direct and indirect jobs, besides employing ~50,000 broiler smallholder families. In addition, 13.5% of the GDP of Brazilian agribusiness in 2022 is a result of the poultry production chain, achieving 21.7 billion dollars [4]” was inserted (Line 40 to 44).

In line 86 there's the expression ‘plays an import role’, which should rather be ‘plays an important role’.

R: The authors agree with the suggestions. The suggested correction was made.

In lines 171 to 174 there is a reference to leisure time but this is not included in Figure 1. It would be better to include it in the figure.

R: The authors agree with the suggestions. The leisure was included in the Figure 1.

Material and Methods

Here the authors should briefly summarise the Emergy model with its variables, relationships and outputs, but unfortunately they fail to do so. There is no explanation of the meaning of the word emergy nor the key concepts in emergy, especially the term transformity. First the authors should explain that the emergy concept evaluates some kind of energy value or energy equivalent of very many factors of a production system. How is information valued, or evaluated as a kind of energy flow ? What is the way of attributing an equivalent energy value to information, to fuel, to feed, to wages, and so on? These concepts should be clearly explained, and then the reader should be presented with a brief description, a kind of summary of what computations will be done step by step. It is not very useful to refer to supplementary material, if the paper is not understandable without this supplementary material. So the materials and methods section should be improved in this sense.

R: The authors agree with the suggestions. To answer “how is the information evaluated as a kind of energy flow” the paragraph “Previous studies used the emergy theory to evaluate the ways and sharing infor-mation as well as its influence on education and culture [18]. However, although such in-formation was a frequent component of computer simulation in emergy synthesis, it was often treated as a peripheral element in larger studies [18].” was inserted to become clearer the evaluation of emergy information assessment in previous studies. (Lines 229 to 232)

To answer “What is the way of attributing an equivalent energy value to information, to fuel, to feed, to wages, and so on”, the sentence “(ii) organisation and elaboration of tables for calculating emergy flows multiplying the transformities by the energy or material flow of each input;” was inserted. (lines 200 to 202)

Also, the paragraph “In this study, the method used to estimate the Tr for cultural information of Santa Ca-tarina State citizens followed an emergy-based model proposed by Odum and Doherty [33] and Odum [25]. The Tr was obtained by dividing the emergy flow of cultural information by the energy storage in Santa Catarina State citizens. The emergy flow of cultural infor-mation was obtained multiplying the emergy flow of the renewable resources of Santa Ca-tarina State, Brazil (in solar emjoules; sej [34]) by 100 years. According to the scientific lit-erature, there is the comprehension that the social interactions among indigenous people suffered over the past ~100 years; the earliest and newest European immigrant people were responsible for promoting the needed knowledge exchange to push and strengthen the broiler industry on the west of Santa Catarina State, making cultural information an important control flow [35,36]. The energy storage in Santa Catarina State citizens was ob-tained considering 10% of human metabolism for social interaction and learned infor-mation (in joules; J) [25]. The populational information of Santa Catarina State, Brazil, for 2018, is according to the Brazilian Institute of Geography and Statistics [37] (Table 1; Sup-plementary Material B, sheet “Cultural information”).” was inserted in order to provide a brief description of what computations were done step by step. (lines 240 to 254)

In line 180 there's a reference to Figure 1 which should rather be Figure 2. Line 182 again refers to Figure 1 the same way as line 171, Please correct it.

R: The authors agree with the suggestions. The suggested correction was made.

In the same line 180 there is the expression ‘ calculation memory’. This is a strange term, shouldn't it be rather ‘calculation method’, or ‘calculation procedure’ or something similar? The same expression occurs in line 293 again.

R: The authors agree with the suggestions. The suggested correction was made.

In section 3.2 the general description of the model is very much missing. The authors refer to Odum [23] for explanation, but this reference is a book, and you can't really expect the reader of the present paper to start to read a full book in order to be able to understand your paper. So please briefly sum up the key concepts, the key terminology, and the key goal of the method you are using.

R: The authors agree with the suggestions. The paragraph “The emergy theory was proposed by Odum [25,30] as being the whole energy previ-ously needed to produce goods and services. The unit of emergy is the solar emjoules (sej), and emergy synthesis considers the effort of nature in human production processes. The methodology depicts the systems features, including their driving forces and interactions [31]. The emergy synthesis applied in this study consist of three steps: (i) elaborating a di-agram of the energy flows of the system, defining the energy sources, the system bounda-ries and the internal components (producers, consumers, stocks, interactions, etc.); (ii) or-ganisation and elaboration of tables for calculating emergy flows multiplying the trans-formities by the energy or material flow of each input; and (iii) calculating the emergy in-dices that support discussion of the system emergy performance. The demand for local re-newable (R) and non-renewable (N) resources are included (I = R+N), as well as inputs from the economy (F) that is divided into renewable (FR) and non-renewable (FN) fractions of each source. The sum of these inputs (Y = I + F) indicates the total emergy demanded by the systems.” was inserted. (lines 194 to 207)

In lines 203 , 205, 208 and 272 there is the expression ‘estimative of ..’ . This is a strange expression, and it should be rather ‘estimation of’, or ‘estimated value of’ as it was mentioned before.

R: The authors agree with the suggestions. The correction were made in the whole text.

In line 215 the last word ‘empower’ should be rather ‘ empowerment’.

R: The authors agree with the suggestions. Actually, the Empower concept (or Maximum Empower Principle) is the Fourth Energy Law proposition which underlies the self-organization of any system. According to Odum, “In the competition among self-organizing processes, network designs that maximize empower will prevail” (Odum, 1996). In the more specific example of ecosystem self-assembly, Odum says, “During the trial and errors of self-organization, species and relations are being selectively reinforced as more energy becomes available to those designs that feed products back into increased production” (Odum, 1988). So, please, consider the term Empower as the most appropriate to this end.

Line 233, Section 3. 2.2: the title uses the word ‘transformity’ assessment, and this concept seems to be a key concept of the model, but its meaning is not explained anywhere.

R: The authors agree with the suggestions. The paragraph “According to Odum [25,30], transformity (Tr) is a measure of the hierarchy of energy, matter or information. As a concept, Tr refers to the energy quality of a given product or service, whilst as an indicator, the Tr refers to the energy conversion efficiency of the production system. For the purpose of this study, the Tr values of the items listed in the calculation table were obtained from the scientific literature and, when necessary, corrected to the. emergy baseline proposed by Brown et al. [32] of 12.0 × ã€–10〗^24 seJ/J.” was inserted to become the concept of Transformity cleaner. (Lines 208 to 213).

Figure 3: what does R mean in the left side of the figure? In the right side of the figure the word ‘foreign’ is mis-spelled as ‘foreing’.

R: The authors agree with the suggestions. The explanation of R was inserted on the footnote “R are the local renewable resources;” (Please see in the Figure 4). Also, the word ‘foreing’ was corrected by ‘foreign’.

In lines 301 and 302 you use the term ‘ producer’. Does this refer to smallholder? Please use the same expression for the same actor.

R: The authors agree with the suggestions. The suggested correction was made.

In line 302 what does ‘sej’ mean?

R: The authors agree with the suggestions. The sentence “The unit of emergy is the solar emjoules (sej), and emergy synthesis considers the effort of nature in human production processes.” was inserted to become the acronym sej cleaner. (lines 195 to 196)

In line 316 you mentioned that ERRP should be equal to 1. Please explain why it is so.

R: The authors agree with the suggestions. Please, found the explanation for both the EER conceptualization and what does the “equal to 1” means in the paragraph: “The model used to estimate a holistic view-based payment for the smallholder broiler family by their services was based on the emergy exchange ratio (EER) indicator as pro-posed by Odum [25]. For Odum, this indicator can help to develop equity in trade, em-ploying shared information among the agents and increasing the benefits for them. This indicator considers the emergy exchanged in a trade or purchase (the given emergy in product form and the received emergy in money form) and is generally expressed con-cerning one trading partner or the other trading partners. Is a measure of the relative trade advantage of one partner over the other [39]. Thus, the best result for EER is 1.0, indicating a balanced trade under emergy units among the trading partners. From  a holistic view, the EER can indicate an imbalanced emergy trade between the buyer (processing firm) and the smallholder broiler family using the buying power of money paid for this end.” (Lines 297 to 307).

Figure 4 is a repetition of figures 1 and 3, and this repetition is unnecessary. As these two figures are slightly modified here, you could explain in a few sentences how figure 1 and figure 3 should be modified to explain the broiler specialties, rather than repeating the same figures again.

R: The authors agree with the suggestions. The Figure 4 (now Figure 5) was improved to meet the suggestion.

Results

The terminology used in the headings of the tables 1-2 and 3 is not explained anywhere.

R: The authors agree with the suggestions. The manuscript was reviwed as a whole and the terminologies was appropriatly included.

In special, the concepts of Tranformity and Emdollar: “; transformity is the ratio obtained by dividing the total emergy that was used in a process by the energy yielded by the process [39]; emdollar is a measure of the money that circulates in an economy as the result of some process and obtained multiplying an emergy flow or storage by the ratio of total emergy to gross domestic product for the national economy [39];” were inserted in each Table footnote.

In addition, the paragraph “In this study, the emergy contribution of each input was allocated to the processing firm (agroindustry) and smallholder according to the allocation approach shown in the Figure 2. This allowed (i) to estimate a payment for the broiler smallholders by their ser-vices by means of a holistic view; and (ii) to estimate the impact of the cultural information from smallholder in the production as a whole. In this section, the discussion is focused on the emergy indicators.” was inserted to become clearer reason to the division between processing firm and smallholder. (Lines 442 to 447)

Instead of giving very extensive footnotes to Table 1, Table 2 and even Table 3, the authors should add a detailed step by step description of how the various numbers in the tables were arrived at.

R: The authors agree with the suggestions. For the authors, the Emergy methodology demands that the footnotes be extensive in order to become the Table/Figures self-explanatory. So, respectfuly, we would like to the reviwer cosider the extensive footnote as a feature of the methodology description.

Please make sure that the abbreviations or acronyms are the same in the text as in the tables. As an example when explaining the content of Table 3 you refer to notations EYR, and ESI, but Table 3 does not have these notations, only the full names, and often in a slightly different wording. Please improve this: use the notations in the table the same way as in the explanation.

R: The authors agree with the suggestions. The acronyms, abbreviations, and notations were verified as a whole. Also, they were inserted in the footnote of the Tables.

Notations within the tables should all be explained, including the measurement unit J (joule?), emdollar, sej, and again, the meaning of ‘transformity’ and of ‘emergy’ are very much missing here.

R: The authors agree with the suggestions. The acronyms, abbreviations, and notations were verified as a whole. Also, they were inserted in the footnote of the Tables.

In Table 1 you use x1013 in the heading, and then further x1011 or x109 in the SOLAR column. It should be better to delete the x1013 from the heading, and then to use in the SOLAR column x1022 or x1024. In the footnote of the table you use the E-notation (e.g. E06) instead of 10 to the correct power (e.g. x 1006 ) . I suggest that you use everywhere the x10 format.

R: The authors agree with the suggestions. The suggested correction was made.

In line 412 and in line 413 the figures are not precise: instead of 4.52 you should have 4.65, and instead of 0.27 you should have 0.26, at least these are the values that you present in Table 3.

R: The authors agree with the suggestions. The suggested correction was made.

The first sentence in line 420 contains percentages about services, feed and labour. How did you arrive to these percentages, where can we see them in tables?

R: The authors agree with the suggestions. To become clearer the origin of services, feed, and labour, the Table 4 was inserted in the manuscript.

In line 428 the last number is 0.72, but there is no such number in Table 3. F

R: The authors agree with the suggestions. Also, to become clearer the comparisons between the results in the manuscript, a column with the indexes results considering the Ci as renewable were inserted in the Table 3.

In the heading of Table 3 it would be better to write out the ‘with CI’ and ‘without CI’ titles.

R: The authors agree with the suggestions. The suggested correction was made.

In Figure 5 please add the actual values to the columns of the figure. Otherwise the explanation of the figure is reasonable, but it would be easier to follow, if the referred numbers are clearly visible in the figure.

R: The authors agree with the suggestions. The suggested correction was made.

Section 4.5 (General insight..) is basically sound and reasonable.

Conclusion

The conclusion section is basically good, though its meaning is somewhat difficult to accept, because of the not satisfactory presentation of the results. Besides, this section should point out the limitations of the present research, and possible continuation of the work.

R: The authors agree with the suggestions. All the “results and discussion” section was improved. Also, a limitation and further studies proposition were inserted: “From an overall perspective,  a methodological approach based on the emergy theo-ry indicates the existing imbalance in monetary exchange between the processing firm and the broiler producer family. The current business model places the responsibility for tan-gible costs (such as infrastructure maintenance and services) on poultry-producing fami-lies, whereas the industry benefits from the intangible costs, i.e. the accumulated cultural knowledge without recognising or compensating it. Introducing a new business/contract model based on the procedure outlined in this paper could push discussions on a more equitable balance between the integrator and the integrated parties. Thus, this model would contemplate more appropriately the importance of the generational acknowledge of broiler production success and then accurately remunerate the contributions of all stake-holders involved.” (Lines 576 to 586)

References

The reference list is quite extensive, containing items from as early as 19 88 and up to date research from the current year. There is a small mistake in item 17 (Abel), In the title of this paper the word ‘emergy’ is spelled with some unnecessary spaces, twice.

R: The authors agree with the suggestions. The suggested correction was made.

Reviewer 4 Report

Comments and Suggestions for Authors

Dear Authors

In their scientific research, the authors relied on the emergy theory. The article is multi-threaded and therefore not fully understandable. In my opinion, it requires simplification and a thorough review of linguistic correctness. In addition to the main content, the article also includes interesting sociological and ethnographic threads. They could be interesting for the reader in a separate article. In the reviewed article, it seems to me that they are unnecessary.

Kind regards

Reviewer

Author Response

General comments

First of all, the authors would like to thank you for your readness in helping us to improve the manuscript. To meet the suggestions of the reviewer, the text was reviewed as a whole. All the comments, suggestions, and observations helped us to improve the study as a whole. We sought to attain all of them expertly. Thus, we hope that the manuscript attains the desired level.

In addition, we inform you that the manuscript has been professionally proofread (please find the English-proof certificate at the bottom of this document).

Reviewer 4

In their scientific research, the authors relied on the emergy theory. The article is multi-threaded and therefore not fully understandable. In my opinion, it requires simplification and a thorough review of linguistic correctness. In addition to the main content, the article also includes interesting sociological and ethnographic threads. They could be interesting for the reader in a separate article. In the reviewed article, it seems to me that they are unnecessary.

Kind regards

R: The authors would like to thank you for your observation. To become the message clearer, the manuscript was improved as a whole. Also, emergy concepts were inserted.

Round 2

Reviewer 2 Report

Comments and Suggestions for Authors

Much improved discussion especially in terms of regional setting and description of model. Expansion of historical background to settlement in Santa Catarina very helpful.

Comments on the Quality of English Language

English improved but still needs considerable editing.

Reviewer 3 Report

Comments and Suggestions for Authors

The authors have very carefully  considered my suggestions and criticisms. The paper has been extensively revised and improved. In the one or two cases when the authors have kept their original presentation instead of conforming to my suggestions, they explained the reasons properly, which I can fully accept.

Therefore I consider the present version of the paper suitable for publication.